# Management of Oral Hygiene in Head-Neck Cancer Patients Undergoing Oncological Surgery and Radiotherapy: A Systematic Review

**DOI:** 10.3390/dj11030083

**Published:** 2023-03-16

**Authors:** Jacopo Lanzetti, Federica Finotti, Maria Savarino, Gianfranco Gassino, Alessandro Dell’Acqua, Francesco M. Erovigni

**Affiliations:** Dental School, Department of Surgical Sciences, University of Turin, 10126 Turin, Italy

**Keywords:** oral hygiene protocol, head-neck cancer, radiotherapy, maxillofacial surgery

## Abstract

Background: In the literature, among oral health prevention programs dedicated to cancer patients, a wide heterogeneity is evident. The purpose of this work is to analyze the available scientific evidence for the treatment of head and neck cancer (HNC) patients undergoing resective surgery and radiotherapy and to draw up a diversified oral hygiene protocol during oncological therapy. Methods: PubMed was used as database. Studies published from 2017 to September 2022 were analyzed. Studies investigating the effectiveness of the preventive procedures carried out by the dental professionals in HNC patients undergoing postoperative adjuvant therapy have been taken into account. Results: The application of the search string on PubMed allowed the selection of 7184 articles. The systematic selection of articles led to the inclusion of 26 articles in this review, including 22 RCTs, 3 observational studies, and 1 controlled clinical study. Articles were divided according to the debated topic: the management of radiation-induced mucositis, xerostomia, the efficacy of an oral infection prevention protocol, and the prevention of radiation-induced caries. Conclusions: Dental hygienists are fundamental figures in the management of patients undergoing oncological surgery of the maxillofacial district. They help the patient prevent and manage the sequelae of oncological therapy, obtaining a clear improvement in the quality of life.

## 1. Introduction

Head-neck cancers (HNC) have a rapid and devastating growth. HNC is one of the most common cancers and a major health problem. The annual incidence of HNC worldwide is approximately 550,000 cases with around 300,000 deaths each year. Usually 90% of all HNC are squamous cell carcinomas. They are mainly loco-regional, and cause serious morphological and functional alterations which, in advanced stages, cause a significant social impact [1].

### 1.1. Introduction to Cancer Treatment, and the Main Problem with Prophylaxis during HNC Treatment

The therapies used for the treatment of these neoplasias are surgery, radiotherapy, and chemotherapy, performed alone or combined. The therapeutic choice depends on the type of tumor, its evolutionary stage, and location. All cancer therapies have undesirable side effects, many involving the oral cavity. Because of this, the treatment plan for patients with HNC includes management and prevention of the side effects of oncological therapy, as well as the maintenance of good oral hygiene to prevent or limit oral complications [2].

Oral prophylaxis performed by oral health professionals is recommended in the pre-surgical phase. In the rehabilitation phase, only minor dental procedures should be performed. This timing aims to minimize the risk of dental and periodontal problems due to the difficulty of practicing oral hygiene after surgery [3].

### 1.2. Radiotherapy

Radiation therapy combined with surgery and/or chemotherapy remains one of the main treatments for both localized and advanced tumors. Most head and neck neoplasms are treated with a total dose between 50 and 70 Gray (Gy), administered in fractions, over a period of 4–7 weeks.

The side effects and toxicities related to radiation therapy can potentially worsen the quality of life of many patients. Despite recent advances in radiation techniques, patients who undergo this therapy in the head and neck district face many dental and oral complications such as oral mucositis, salivary gland dysfunction, radiation caries, and osteoradionecrosis [4].

### 1.3. Surgery

Advanced stage lesions treated with surgery can cause numerous aesthetic, functional, and psychological sequelae [4]. The surgical excision of tumors in the maxilla is a reason for a surgery called maxillectomy or a maxillary resection surgery. It depends on the type and location of the lesion; cancer ablation surgery of the maxilla often involves the hard palate, maxillary sinus, and nasal cavity.

Before the surgical phase, dental prophylaxis is recommended. Minor operative dental procedures should be performed, with the purpose of minimizing the risk of dental and periodontal problems due to the difficulty of oral hygiene practice post surgically.

### 1.4. Management

In order to reduce the severity of side effects associated with radiotherapy, regular follow-ups need to be arranged and these patients need to be monitored. The oral hygiene of patients should be optimized through oral health education, modification of risk factors (including xerostomia and altered nutritional requirements), caries prevention regimens, and fluoroprophylaxis [5].

It is necessary to intervene with a multidisciplinary approach, especially in patients undergoing resective surgery and radiotherapy of the head and neck region, placing the patient at the center of a team that interacts for their well-being and therapeutic success. In this context, maxillofacial prosthesis is an essential component of the oral rehabilitation of patients with oral cancer undergoing surgical exeresis. The main goals of the maxillofacial prosthesis are to restore oral function and to improve the patient’s facial aesthetics and quality of life [3].

### 1.5. Role of Dental Health Professionals

The goals of oral hygiene management in patients with HNC varies depending on the cancer treatment. The role of the dental hygienist includes oral screening and management of oro-dental complications due to surgery, radiation, and/or chemotherapy [6].

Dental hygienists perform maintenance of good oral hygiene, which is fundamental in preventing or limiting painful episodes or dangerous infections. With the goal of improving the quality of life by promoting personal care, the dental hygienist plays a significant role in the management of cancer patients by helping them to achieve good oral hygiene, removing tartar deposits that can be a cause of infection, maintaining trophic oral mucosa, educating to manage treatment-related problems, and providing the patient with the means for prevention.

There is little scientific information in the literature about the oral hygiene of cancer patients who have to undergo maxillary resection surgery, have been maxillectomized, or wear a prosthetic obturator.

In the literature, wide heterogeneity is evident in oral health prevention programs dedicated to cancer patients, making it difficult to draw conclusions about the superiority of some protocols over others [7].

The purpose of this paper is to synthesize the available evidence for the treatment of the head and neck cancer patient undergoing resective surgery and radiotherapy, with or without concomitant chemotherapy, and to introduce the diversified oral hygiene protocol in the various stages of cancer therapy used in S.C. Oral Rehabilitation Maxillofacial Prosthetics Dental Implantology of the C.I.R. Dental School in Turin.

## 2. Materials and Methods

The systematic review relied on the Preferred Reporting Items for Systematic Reviews and Meta-Analyses (PRISMA) statement with the use of the PICO (Population, Intervention, Comparison, Outcome) tool in order to structure the search questions [8,9].

Studies were selected for the review following the PICOS criteria as follows: *Participants* Patients diagnosed HNC and undergoing cancer ablation surgery and postoperative adjuvant RT with or without CCRT.*Intervention* Professional and home oral hygiene techniques.*Comparator* Comparison with different professional and home oral hygiene techniques.*Outcome* Improvement of the condition of the patient undergoing RT.

Studies published up to September 2022 (included) and in the previous 5 years were analyzed to evaluate the recent literature on this topic. The review includes studies written in English or Italian. Studies investigating the effectiveness of the preventive procedures of the dental hygienist in HNC patients undergoing postoperative adjuvant RT with or without CCRT have been taken into account.

A further search was performed after drafting the article in February 2023.

PubMed was used as database. The search strategy was based on the following key words, in multiple combinations, that were chosen to reflect the focus of the review: “head and neck neoplasm”, “maxillofacial prostheses”, ”maxillofacial surgery”, “chlorhexidine”, “dental hygienist”, “home oral care”, “home oral hygiene”, “dental brush”, “toothbrush”, “mouth rinse”, ”oral hygiene”, “toothpaste”, “fluoride”. The search equations were ‘(head and neck neoplasm OR maxillofacial prostheses OR maxillofacial surgery) AND (chlorhexine OR dental hygienist OR home oral care OR home oral hygiene OR dental brush OR tooth brush OR mouth rinse OR oral hygiene OR tooth paste OR fluoride)’.

A qualitative assessment has been carried out based on adherence to the eligibility criteria, the completeness of the description of the methodology, and the study design. 

Two authors (L.J. and S.M.) were involved in the literature search. The choice of reference studies was made primarily by filtering for the year of publication, language, and type of study, and secondly through the evaluation of the abstracts and full-text of the articles, in a non-blind but independent process. 

The independent lists were cross-referenced; any disagreement was resolved by consensus or with a third-party reviewer (F.F.). Then, in line with inclusion and exclusion criteria, a full-text eligibility assessment was performed by the two reviewers in a blinded procedure, after which the process of referencing and citation searching was made. A 100% agreement rate was obtained between the two authors.

The following data were collected: topic, author’s name, year of publication, study design, aim of the study, sample size, and conclusions. A standardized form was used to extract data from the included studies.

## 3. Results

The application of the search string on PubMed resulted in the selection of 7186 results, of which articles published before 2017, or not published in English or Italian were discarded. Of the remaining papers, only randomized and non-randomized controlled trials (RCTs) and observational studies were considered, obtaining 221 articles. Then, a selection was performed on the basis of the abstract, cutting down to 33 articles.

Another seven papers were excluded after reading the full-text because they were not relevant to the topic of the review: three of these did not consider patients undergoing resective oncologic surgery and/or radiotherapy, and another four papers did not present maneuvers within the dental hygienist’s expertise.

Figure 1 shows the eligibility screening steps and Table 1 shows the details of the selected studies.

### 3.1. Radio-Induced Mucositis

Fourteen studies included are RCTs and one is an observational study. 

The articles have evaluated the effectiveness of different types of mouthwashes with the aim of delaying the onset or reducing the severity of mucositis induced by radiotherapy performed in the head and neck district. 

Huang et al. [10] shows that the use of a saline-based mouthwash and an oral health education protocol performed during radiotherapy lead to an increase in the quality of life of the radio-treated patient, even if these interventions do not affect the symptomatology or the onset of mucositis. In addition, the double-blind RCT by Manifar S. et al. [24] showed the significant reduction in oral mucositis intensity due to a symbiotic (probiotic) mouthwash.

The reduction in pain symptomatology during radiotherapy due to the use of doxepin mouthwash or diphenhydramine–lidocaine–antacid mouthwash is also demonstrated by Sio et al. [20].

The multicenter study by Lalla et al. [14] demonstrates the efficacy in the prevention of radio-induced mucositis of a mouthwash based on hydrogen peroxide, eugenol, camphor, and parchlorophenol. Lozano et al. [19] tested the efficacy of a high-dose (3%) melatonin mucoadhesive oral gel, demonstrating decreased incidence of severe mucositis and duration of ulcerative mucositis. Y. Jun et al. [23] showed the efficacy of an antiulcer oral mucosal protectant mouthwash in the incidence and severity of radio-induced mucositis and in improving the quality of life.

The prospective observational study by Morais et al. [22] draws attention to the effectiveness of a prevention protocol involving daily oral hygiene control, removal of possible infectious foci, fluoroprophylaxis, and a daily session of photobiomodulation therapy with an idium gallium arsenic phosphate diode laser, concluding that the combination of oral prevention and photobiostimulation is effective in preventing the immediate adverse effects of radiotherapy (oral mucositis, pain, dysphagia, xerostomia) and in improving the quality of life.

Several studies compare the effectiveness of mouthwashes based on plant extracts and natural substances with a placebo, saline, or baking soda-based mouthwash. The effectiveness of these mouthwashes is often comparable with bicarbonate-based saline solution rinses.

Aghamohammadi et al. [12] demonstrated that a mouthwash based on Zataria extracts affected the incidence of grade 3–4 oral mucositis to a relative risk ratio of 0.432 and also had a significant effect on the patient’s pain symptomatology.

In contrast, payayor and fingerroot mouthwashes analyzed by Kongwattanakul et al. [11], Cystus tea mouthwash used by Ebert et al. [15], green tea mouthwash used by Liao et al. [16], propolis mouthwash by Hamzah et al. [18], and curcumin mouthwash studied by Shah et al. [21] have been shown to be equally effective compared to placebo mouthwash, especially when assisted with cryotherapy and laser or light therapy [16]. The multicenter study by De Sanctis et al. [17] also found no differences in the onset and severity of radio-induced mucositis with Lactobacillus brevis CD2 supplementation.

In addition, the efficacy of a granulocyte macrophage colony stimulating factor mouthwash was evaluated by Marylin et al. [13], obtaining no statistically significant results. However, the authors underline the importance of subjecting the radio-treated patient to regular assessments, education, coaching, and follow-up during the period of cancer therapy.

### 3.2. Prevention or Management of Xerostomia Due to Radiation Therapy

Six RCTs evaluating the efficacy of different protocols were selected.

Marimuthu et al. [30] evaluated the efficacy of an immunologically active saliva substitute mouthwash based on the *Shortened Xerostomia Inventory and Unstimulated Whole Saliva*, verifying the validity of the presidium. The study by Sio et al. [27] also reports the potential benefits of N-acetylcysteine rinses in radio-chemotherapy-treated patients.

Lam-ubol et al. [29] and Nuchit et al. [25] compared a particular salivary substitute gel, oral moisturizing jelly, with a commercial salivary substitute, demonstrating that both products are effective in improving the xerostomia condition of the radio-treated patient, especially when used continuously for at least one month after radiotherapy. Apperley et al. [28] tested a salivary substitute based on methylcellulose, concluding that there was no difference between the tested gel and placebo.

In his work, Jiang et al. [26] applies a supportive protocol for patients undergoing radiotherapy of the head-neck district, consisting of home oral hygiene education sessions, advice on cessation of unhealthy lifestyles (smoking, dietary advice), and instructions to stimulate facial and tongue muscles with the aim of preserving salivary gland function and salivary flow, achieving, in the first year after radiation therapy, a significant improvement in the condition of xerostomia and an increase in unstimulated saliva.

### 3.3. Prevention of Postoperative Infection and Oral Health Improvement

One RCT study and two observational studies were selected.

The retrospective observational studies by Gondo et al. [34] and Ishimaru et al. [33] note the importance of including the patient who is to undergo resective oncologic surgery in oral care programs before and after surgery, in order to reduce postoperative infections. In addition, Ishimaru et al. [33] states that patient preparation through oral prevention pathways is associated with a significant decrease in postoperative pneumonia and all-cause 30-day mortality following cancer resection.

The only selected RCT study on the topic, by Sohn et al. [35], reiterates the importance of scheduling regular dental visits, oral prevention, and giving home oral hygiene instructions to patients during radiation therapy.

### 3.4. Radio-Induced Caries Prevention

One RCT and one controlled clinical study were selected. The article by Sim et al. [32] shows that treatment with biomimetic CPP-ACP saliva together with SnF2/NaF significantly reduces caries progression in patients undergoing radiotherapy in the head-neck district. Furthermore, the controlled clinical study by Lee et al. [31] suggests that preventive fluoride applications show positive changes in oral health of the radio-treated patient in the head-neck district involving caries, plaque index, and gingival inflammation.

### 3.5. Risk of Bias

The risk of bias of the individual studies is reported in Table 2. None of the included studies were judged at low risk of bias for all domains. Eight studies were judged at high risk of bias (M.J. Dodd (2022) [13]; N. Ebert et al. (2021) [15]; M.H. Hamzah et al. (2022) [18]; A. Lozano et al. (2021) [19]; S. Shah et al. (2020) [21]; Y. Jun et al. (2022) [23]; O. Apperley et al. (2017) [28]; A. Lam-ubol et al. (2021) [29]; H.O. Sohn et al. (2018) [35]).

Three of the four analytical studies included earned the maximum score (Table 3).

## 4. Discussion

The purpose of this review was to identify the available scientific evidence for the treatment of the head and neck cancer patient undergoing resective surgery and radiotherapy, with or without concomitant chemotherapy.

As previously mentioned, the dental professional has a significant role in the management of the oncological patient and plays a fundamental role in preventing or limiting painful episodes or dangerous infections caused by surgery and radiotherapy. Furthermore, we described a recommendation of operational protocols for dental professionals based on experience gained from S.C. Oral Rehabilitation Maxillofacial Prosthetics Dental Implantology of the C.I.R. Dental School in Turin, (Figure 2).

In the preparation prior to maxillary resection surgery, it is necessary to assess the patient’s oral health and hygiene status.

The prevention, hygiene, and maintenance program must be individualized and flexible in order to achieve a continuous and progressive educational process. Professional and home oral hygiene with a related instruction prior to maxillofacial surgery contributes to the oral health and well-being of the patient, as well as significantly reducing pneumonia and mortality [36,37].

In the post-surgical phase, the patient should perform home oral hygiene using the following aids:-Small-headed soft-bristled toothbrush, using an atraumatic brushing technique [38];-Fluoride toothpaste [38,39];-Topical application of fluoride gel with special trays one time a day for 5 min (two times a day during radiotherapy) [38,39];-Alcohol-free 0.12% chlorhexidine rinses [38,39];-Saline solution rinses;-Interdental hygiene with dental floss or interdental brush [38];-Tongue cleansing with soft toothbrush or gauze [38].

If the patient is also treated with radiotherapy of the head-neck district, the patient’s oral cavity should be cleansed, with any periodontal causal therapy treatments at least 15 days before the beginning of radiation therapy to avoid biological complications and reduce the risk of osteoradionecrosis [36]. 

Radio-induced oral mucositis affected the oral mucosa by direct and indirect radiation damage and is characterized by rapid atrophy that can lead to erosive lesions. This condition can be complicated by fungal overinfection and can cause pain, difficulty in feeding and swallowing. Radiotherapy of the head-neck district causes a decrease in salivary flow, which implies a substantial change in the homeostasis of the oral cavity and may result in a condition of xerostomia characterized by dysgeusia and difficulty in swallowing.

Professional and at home oral hygiene can improve the patient’s mucositis condition [40].

The use of antiseptic mouthwashes alone may have utility in improving oral hygiene and preventing possible overinfections [40].

Mouthwashes with doxepin and diphenhydramine–lidocaine–antacid (DLA) are not effective in the prophylaxis of mucositis, but they are effective in relieving pain due to this condition, while granulocyte macrophage colony stimulating factor mouthwashes are the most studied mouthwashes for the prevention of mucositis [40]. 

Therefore, daily oral hygiene, rinses with saline solution mouthwashes, in combination with laser therapy, are very effective in the treatment of oral mucositis induced by radiotherapy and chemotherapy [41]. 

Low Level Laser Therapy is used with an exposure time of 15 min, in which 180 Joules are delivered. The tip the laser beam comes out of is kept 2 mm away from the target tissue. There is a tendency to concentrate the entire 15 min treatment in the parts of mucosa that show the most inflammation or mucositis lesions. In cases where there are no particularly significant mucositis or inflammation situations, treatment is carried out with a preventive purpose on the entire oral cavity: genial mucosa, soft palate, lingual margins, and oral floor. In such cases, from the point of view of prevention, the patient is advised to hydrate continuously and consistently and use mucin-based salivary substitutes and xylitol chewing gums. Artificial saliva tends to have a long duration of action and differs in formulation (e.g., spray, gel), pH, lubricant (e.g., mucin, carboxymethyl cellulose), and other ingredients (e.g., flavoring, fluoride). Ideally, artificial saliva should have a neutral pH and contain fluoride [38].

Radio-induced caries are a late side effect that can occur about three months after treatment. They are mainly localized at the level of the dental neck and are related to reduced salivary flow, change in saliva quality, and pH reduction. Therefore, in addition to scrupulous oral hygiene, topical fluoroprophylaxis using individual trays (daily 5 min applications of 1.1% NaF gel or Amorphous Calcium Phosphate) [42] is recommended.

The maintenance program and professional oral hygiene follow-ups are critical to maintaining the patient’s long-term oral health. The maintenance program includes oral hygiene motivation and instruction, an oral mucosal health status check, and professional oral hygiene every 3 months.

In the case of patients with removable dentures or a palatal obturator, the prosthetic restoration is brushed with a denture brush and Marseille soap, a hard soap made from vegetable oils.

Marseille soap and disinfecting it by soaking it daily in a solution with water and 2% sodium hypochlorite or chlorhexidine is recommended [38].

The present systematic review has limitations. First, only one database was consulted, and the search was restricted to the English and Italian languages. Owing to the diversity of the included study concerning sample size, administration, and duration of the intervention, radiation dose and technique, and assessment methods, a quantitative analysis of all the studies was not possible. Moreover, the research protocol has not been registered on any of the current databases for systematic reviews.

## 5. Conclusions

Patients undergoing maxillofacial district surgery for oncological conditions face a very complex set of issues due to the many implications of disease diagnosis, treatment, and rehabilitation.

The range of disability varies from minimal impairment to severe functional impairment. Oral health professionals caring for these patients need to know what sequelae will result from the treatments, and what therapeutic changes can significantly improve the rehabilitation process. To achieve this and provide the patient with the most appropriate care, all specialists involved in rehabilitation must discuss with each other and integrate their respective knowledge for this purpose.

For the management of this type of patient, the dental hygienist is a key figure that can prepare them before and follow them during and after surgery, accompanying them with regard to post-surgical, radio and chemotherapy sequelae, helping them to achieve a remarkable improvement in their quality of life.

## Figures and Tables

**Figure 1 dentistry-11-00083-f001:**
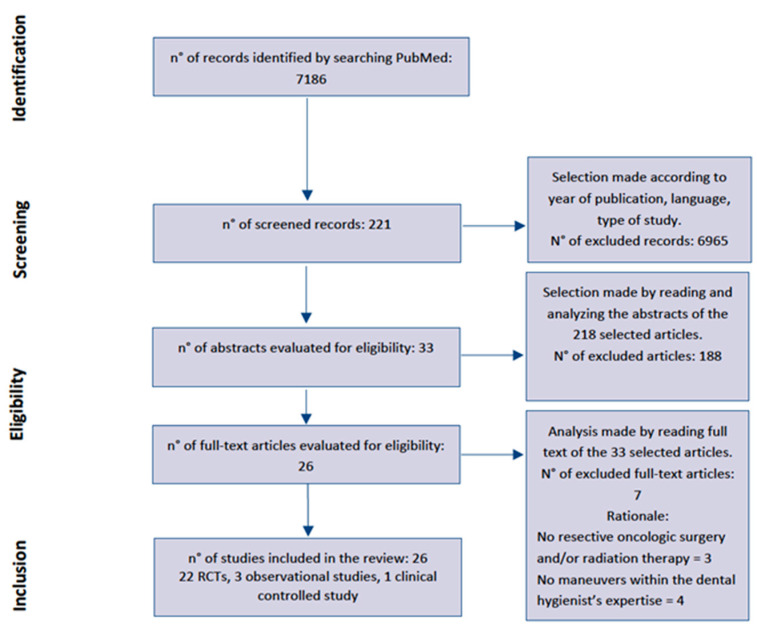
Flowchart of the systematic review process.

**Figure 2 dentistry-11-00083-f002:**
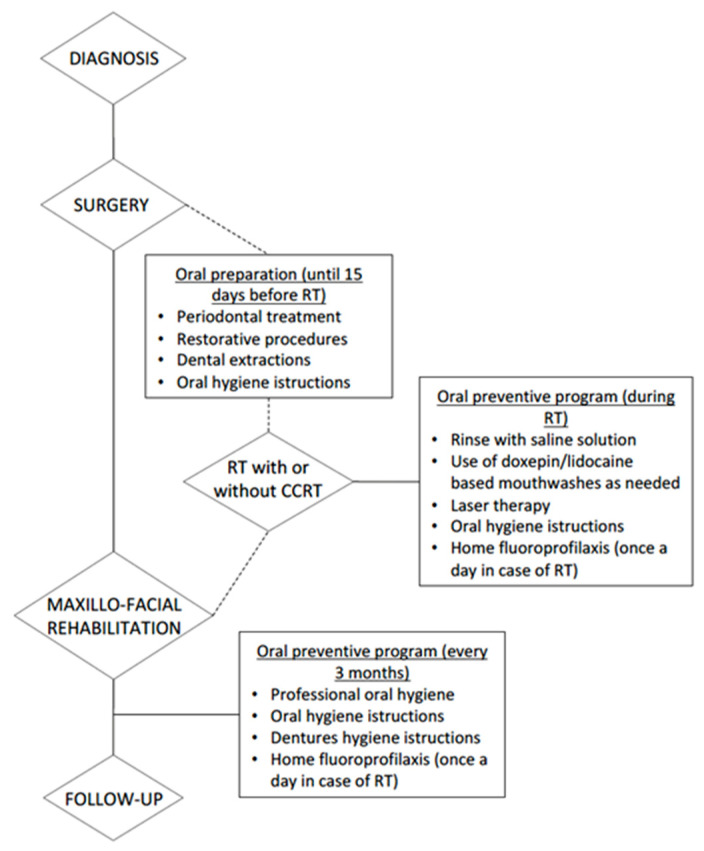
Flowchart of recommended operational protocol for oral health management in HNC patients (RT: radiation therapy; CCRT: concurrent chemotherapy).

**Table 1 dentistry-11-00083-t001:** Systematic reviews.

Topic	Authors	Study Design	Aim of the Study	Sample	Conclusions
Oral mucositis	B.-S. Huang et al. (2018) [10]	RCT with two groups	To analyze the effectiveness of a saline mouthrinse and the efficacy of an education programme	91 patients diagnosed oral cavity cancer and undergoing postoperative adjuvant RT and CCRT	Saline mouth rinses together with an education programme were found effective in increasing physical and social-emotional QOL by improving the radiation-induced OM symptoms and promoting oral comfort
Oral mucositis	S. Kongwattanakul et al. (2022) [11]	RCT with three groups	To compare the efficacy of two mouthwashes containing extracts of payayor and fingerroot with the standard care of saline solution (0.9%) with sodium bicarbonate in preventing OM in head and neck cancer patients receiving radiotherapy	121 patients with HNC undergoing RT	Payayor and fingerroot mouthwashes were found as having the same efficacy in reducing the severity of OM and could cause a slight delay in the onset of the symptoms
Oral mucositis	A. Aghamohammadi et al. (2018) [12]	Double-blind RCT	To assess the efficacy of Zataria Multiflora mouthwash in reducing the incidence of OM in HNC patients undergoing radiotherapy	52 patients with HNC undergoing RT	After 6 weeks of treatment with Zataria Multiflora mouthwash, OM affected significantly less patients than in the placebo group
Oral mucositis	M.J. Dodd (2022) [13]	Double-blind RCT	To compare the effectiveness of the granulocyte macrophage colony stimulating factor mouthwash to Salt and Soda mouthwash for both the prevention (prior to onset of OM) and treatment (from onset of OM to its healing) of OM	91 patients with HNC receiving a cumulative dose of RT of between 5500 and 7000 cGy (cGys) over 6–7 weeks	The granulocyte macrophage colony stimulating factor mouthwash was not found to be more effective than Salt and Soda mouthwash neither in the prevention nor in the treatment of OM
Oral mucositis	R.V. Lalla et al. (2020) [14]	Parallel-group, double-blind RCT	To assess the efficacy of Dentoxol^®^ in reducing the severity of OM	108 patients with HNC undergoing RT at least 5000 cGy, with or without CCRT	Dentoxol used 5 times/day caused significantly fewer time-points with severe OM and caused a delay in the onset of severe OM, compared with a control rinse
Oral mucositis	N. Ebert et al. (2021) [15]	Prospective, single-center, randomised phase III trial	To compare Cystus^®^ tea effects as mouthwash to sage tea effects on OM in patients undergoing RT and CCRT for HNC	57 patients with histologically confirmed HNC	Cystus^®^ and sage tea effects were found not statistically different. This mouthwash can be applied in addition to accurate oral care and hygiene along with the application of fluorides
Oral mucositis	Y.C. Liao et al. (2021) [16]	Prospective, single-blind RCT	To evaluate the effectiveness of green tea mouthwash in the improvement of oral health in oral cancer patients undergoing cancer treatment	61 HNC patients treated with oral surgery and RT with and without CCRT	The prolonged use of green tea mouthwash was effective in improving and mantaining the oral health status
Oral mucositis	V. De Sanctis et al. (2019) [17]	Multicentric, phase III, open-label RCT	To investigate the effects of *Lactobacillus Brevis* CD2 in preventing OM onset during RT	68 patients with histologically diagnosed HNC, except larynx, parotid and other salivary glands under RT and CCRT	*Lactobacillus Brevis* CD2 lozenges were not demonstrated to be effective in preventing radiation-induced mucositis in patients with HNC
Oral mucositis	M.H. Hamzah et al. (2022) [18]	Prospective, double-arm, RCT with intervention	To evaluate the effects of a 2.5% propolis mouthwash in the prevention of RT-induced OM in patients with nasopharyngeal carcinoma	17 patients with nasopharyngeal carcinoma who underwent HNC surgery	Propolis mouthwash was found statistically effective in reducing the severity of OM following RT
Oral mucositis	A. Lozano et al. (2021) [19]	Multicentric, phase IIa, prospective, double-blind RCT	To evaluate the effectiveness of melatonin oral gel mouthwashes in preventing and treating OM in patients in treatment for HNC	79 patients with HNC treated RT and CCRT	3% melatonin oral gel caused a lower incidence and a shorter duration of OM
Oral mucositis	T.T. Sio et al. (2019) [20]	3-group, double-blind, phase 3, RCT	To assess the efficacy of doxepin mouthwash or diphenhydramine–lidocaine–antacid mouthwash on OM-related pain	230 patients with HNC treated RT	Patients undergoing head and neck radiotherapy, and using doxepin mouthwash or diphenhydramine–lidocaine–antacid mouthwash compared with placebo showed a significant reduction in OM-related pain during the first 4 h after administration
Oral mucositis	S. Shah et al. (2020) [21]	Parallel arm, triple-blinded RCT	To compare the effectiveness of 0.1% curcumin and 0.15% benzydamine mouthwash on radio-induced OM	74 patients with histological confirmation of HNC, scheduled to receive RT	0.1% curcumin mouthwash significantly delayed the onset of OM
Oral mucositis	M.O. Morais et al. (2020) [22]	Prospective observational study	To assess oral complications and quality of life in HNC patients undergoing a preventive oral care program and photobiomodulation therapy	61 patients diagnosed with HNC undergoing RT and CCRT	The photobiomodulation therapy reduced quality of life impacts and interruption of RT due to severe OM
Oral mucositis	Y. Jun et al. (2022) [23]	RCT	To assess the efficacy and safety RADoralex^®^ in preventing and treating radiation-induced oral mucosal reactions	90 patients with locally advanced Nasopharyngeal carcinoma who received RT and CCRT and showed post-treatment grade 1 oral mucositis	OM incidence and severity was reduced and the progression was delayed using RADoralex^®^
Oral mucositis	S. Manifar et al. (2023) [24]	Double-blind RCT	To assess and compare the effects of a synbiotic mouthwash with a saline mouthwash on preventing and controlling radiotherapy-induced OM in oral cancer patients	64 patients with oral cancer, who received 6000 cGY of RT in 34 fractions	Synbiotic mouthwash caused a significant reduction in OM intensity and prevented its onset in oral cancer patients undergoing RT
Xerostomia	S. Nuchit et al. (2020) [25]	Single-blinded RCT	To evaluate the effectiveness of an edible saliva substitute, on dry mouth, swallowing ability, and nutritional status in post-RT HNC patients	62 patients with HNC who have completed RT at least 1 month earlier	Using saliva substitutes (OMJ or GC) continuously for at least a month improved dry mouth condition and enhanced swallowing ability
Xerostomia	N. Jiang et al. (2022) [26]	RCT	To investigate the effects of an integrated supportive program on xerostomia and saliva characteristics in patients with HNC 1 year post-RT	92 patients with histologically diagnosis of HNC who received a low dose RT to the major salivary glands	Patients with HNC with a low dose RT to the major salivary glands who were followed up for 12 months post-RT in an integrated supportive program with good adherence experienced a relief in xerostomia and an increase in unstimulated saliva flow rate
Xerostomia	T.T. Sio et al. (2019) [27]	Prospective, double-blind RCT	To evaluate the effectiveness of N-acetylcysteine rinse in improving thickened secretions and dry mouth during and after RT	32 patients undergoing CCRT for HNC	After using N-acetylcysteine rinse patients reported an improvement in thickened saliva and xerostomia
Xerostomia	O. Apperley et al. (2017) [28]	RCT	To compare a novel oily emulsion to a currently available saliva substitute	29 patients with xerostomia after RT to the HNC	Patients reported no clinically significant difference between the novel oily formulation, methylcellulose, and water
Xerostomia	A. Lam-ubol et al. (2021) [29]	Single-blind RCT	To assess the effectiveness of an edible artificial saliva gel, an oral moisturizing jelly, and a topical commercial gel on Candida colonization and saliva properties	56 post-RT HNC patients with xerostomia	Both the gels reduced the number of Candida species and improved saliva properties in post-RT patients with xerostomia
Xerostomia	D. Marimuthu et al. (2021) [30]	Prospective, double-blind RCT	To investigate the effects of a saliva substitute mouthwash in patients with nasopharyngeal cancer and xerostomia	94 patients patients diagnosed with nasopharyngeal carcinoma who underwent either RT or CCRT	Xerostomia scores were reduced and salivary flow improved after using an immunologically-active saliva substitute mouthwash
Radio-induced caries	H.J Lee et al. (2021) [31]	Quasi-experimental, non-synchronized non-equivalent case-control study	To investigate the effect of the comprehensive oral care program on oral health status and symptoms in HNC patients undergoing RT	61 Patients undergoing RT for non-metastatic HNC	Comprehensive oral care intervention is effective in preventing dental caries and increasing the quality of life in HNC patients
Radio-induced caries	C.P.C. Sim et al. (2019) [32]	Double-blind RCT	To evaluate the effects of treatment with the saliva biomimetic, casein phosphopeptide-amorphous calcium phosphate and SnF2/NaF compared with SnF2/NaF alone on the progression of coronal surface caries in HNC patients undergoing RT	24 patients (2685 tooth surfaces) undergoing RT to the HNC	The progression of coronal surface caries was reduced by the use of the saliva biomimetic, casein phosphopeptide-amorphous calcium phosphate, and SnF2/NaF
Preventive oral care protocol	M. Ishimaru et al. (2018) [33]	Retrospective observational cohort study	To investigate the association between preoperative oral care and postoperative complications in patients undergoing major HNC surgery	509,179 patients who underwent surgery for HNC and other cancers	The preoperative oral care, in patients with HNC, led to a significant decrease in postoperative pneumonia and all-cause 30-day mortality
Preventive oral care protocol	T. Gondo et al. (2020) [34]	Retrospective observational study	To evaluate the incidence of postoperative pneumonia and surgical site infection in HNC patients and investigate the link between oral care and postoperative infection	209 patients who underwent HNC surgery	Oral care before and after surgery reduced postoperative infections in patients with HNC
Preventive oral care protocol	H.O. Sohn et al. (2018) [35]	RCT	To investigate the effects of professional oral hygiene care during RT in patients with HNC	27 patients with HNC who underwent RT	Periodic dental visits, oral hygiene care, and instructions improved oral health in patients with HNC during RT

RT: radiation therapy; CCRT: concurrent chemotherapy; OM: oral mucositis; HNC: head and neck cancer.

**Table 2 dentistry-11-00083-t002:** Summary of risk of bias in individual interventions studies.

Authors (years)	Study’s Design	Randomization	Allocation Concealment	Assessor Blinding	Operators Blinding	Missing Outcome Data Reported	Missing Outcome Were Balanced among Groups	Reasons for Drop out	Selective Outcome Reporting	Statistical Method	Sample Size Estimation	Examiner Calibration
B.-S. Huang et al. (2018) [10]	RCT	Adequate	Adequate	NR	NR	Adequate	Adequate	Adequate	Adequate	Adequate	NR	NR
S. Kongwattanakul et al. (2022) [11]	RCT	Adequate	Adequate	NR	NR	Adequate	Adequate	Adequate	Adequate	Adequate	Adequate	NR
A. Aghamohammadi et al. (2018) [12]	RCT	Adequate	Adequate	Adequate	Adequate	Adequate	Adequate	Adequate	Adequate	Adequate	NR	NR
M.J. Dodd (2022) [13]	RCT	Unclear	Unclear	Adequate	Adequate	Adequate	Adequate	Adequate	Adequate	Adequate	NR	Adequate
R.V. Lalla et al. (2020) [14]	RCT	Adequate	Adequate	Adequate	Adequate	Adequate	Adequate	Adequate	Adequate	Adequate	Adequate	Adequate
N. Ebert et al. (2021) [15]	RCT	Unclear	NR	NR	NR	Adequate	Adequate	Adequate	Adequate	Adequate	Unclear	NR
Y.C. Liao et al. (2021) [16]	RCT	Adequate	Adequate	NR	NR	Adequate	Adequate	Adequate	Adequate	Adequate	Adequate	NR
V. De Sanctis et al. (2019) [17]	RCT	Adequate	Adequate	NR	NR	Adequate	Adequate	Adequate	Adequate	Adequate	NR	NR
M.H. Hamzah et al. (2022) [18]	RCT	NR	NR	NR	NR	Inadequate	Inadequate	NR	Adequate	NR	NR	NR
A. Lozano et al. (2021) [19]	RCT	Unclear	Adequate	NR	NR	Adequate	Adequate	Adequate	Adequate	Adequate	Adequate	NR
T.T. Sio et al. (2019) [20]	RCT	Adequate	Adequate	Unclear	Unclear	Adequate	Adequate	Adequate	Adequate	Adequate	Adequate	NR
S. Shah et al. (2020) [21]	RCT	Unclear	Unclear	Adequate	Adequate	Adequate	Adequate	Adequate	Adequate	Adequate	Adequate	Adequate
Y. Jun et al. (2022) [23]	RCT	Unclear	Adequate	NR	NR	Adequate	Adequate	Adequate	Adequate	Adequate	Adequate	NR
S. Manifar et al. (2023) [24]	RCT	Adequate	Adequate	Adequate	Adequate	Adequate	Adequate	Adequate	Adequate	Adequate	NR	NR
S. Nuchit et al. (2020) [25]	RCT	Adequate	Adequate	Adequate	Adequate	Adequate	Adequate	Adequate	Adequate	Adequate	Adequate	NR
N. Jiang et al. (2022) [26]	RCT	Adequate	Adequate	NR	NR	Adequate	Adequate	Adequate	Adequate	Adequate	Inadequate	NR
T.T. Sio et al. (2019) [27]	RCT	Adequate	Adequate	Unclear	Unclear	Adequate	Adequate	Adequate	Adequate	Adequate	NR	NR
O. Apperley et al. (2017) [28]	RCT	Unclear	Adequate	NR	NR	Adequate	Adequate	Adequate	Adequate	Adequate	Adequate	NR
A. Lam-ubol et al. (2021) [29]	RCT	Unclear	Adequate	NR	NR	Adequate	Adequate	Adequate	Adequate	Adequate	Adequate	NR
D. Marimuthu et al. (2021) [30]	RCT	Adequate	Adequate	NR	NR	Adequate	Adequate	Adequate	Adequate	Adequate	Adequate	Adequate
C.P.C. Sim et al. (2019) [32]	RCT	Adequate	Adequate	Unclear	Unclear	Adequate	Adequate	Adequate	Adequate	Adequate	Adequate	NR
H.O. Sohn et al. (2018) [35]	RCT	Unclear	Unclear	NR	NR	Inadequate	Inadequate	Inadequate	Adequate	Adequate	NR	NR

RCT: randomized controlled trial; NR: not reported; CT: controlled trial.

**Table 3 dentistry-11-00083-t003:** Summary of risk of bias in analytical studies.

Authors (years)	Study’s Design	Representativeness of the Exposed Subjects	Selection of Non-Exposed Subjects	Ascertainment of Exposure	Ascertainment of Outcome	Comparability of Exposed and Non-Exposed Groups on the Basis of the Design or Analysis	Assessment of Outcome
M.O. Morais et al. (2020) [22]	Prospective observational study	Adequate	Inadequate	Adequate	Adequate	Adequate	Adequate
H.J Lee et al. (2021) [31]	Case-control study	Adequate	Adequate	Adequate	Adequate	Adequate	Adequate
M. Ishimaru et al. (2018) [33]	Retrospective observational cohort study	Adequate	Adequate	Adequate	Adequate	Adequate	Adequate
T. Gondo et al. (2020) [34]	Retrospective observational study	Adequate	Inadequate	Adequate	Adequate	Adequate	Adequate

## Data Availability

Data are available from the corresponding authors upon reasonable request.

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
