# Peer review of "Management of Oral Hygiene in Head-Neck Cancer Patients Undergoing Oncological Surgery and Radiotherapy: A Systematic Review"

_dentistry, 2023, doi:10.3390/dj11030083_

Round 1

Reviewer 1 Report

The manuscript summarizes the results of recently published available evidence on the oral care of HNC patients. The topic has high importance, and the results are widely applicable, although the paper in its recent form requires a major improvement. The paper will benefit from proofreading by a native English speaker.

Abstract: 

1.     First sentence - please add oral health before “prevention programs.”

2.     Ln25 - preventive procedures are not performed only by dental hygienists. “Dental hygienist” should be substituted by dental professionals.

3.     Ln30 – “oral prevention protocols” should be changed for infection prevention protocols

Introduction:

1.     Adding HNC prevalence data (international aspect) at the beginning will strengthen the importance of the aims of this paper.

2.     Please reorganize the manuscript into paragraphs. A few sentences are grouped and abruptly separated with indents throughout the manuscript, breaking the reading flow. Use five paragraphs in the introduction, collecting the following information into parts 1. introduction of cancer treatment, state the main problem with prophylaxis during HNC treatment, 2. radiotherapy, 3. surgery, 4. management, 5. role of dental health professionals.

3.     Ln 59-72 should be moved before “Advanced stage lesions “in Ln47

4.     Ln51-54 has no relevance to the present review, please remove it

5.     Ln55-58 should be moved after Ln75

6.     Dental hygienists in Ln55 should be changed to oral health professionals

7.     Ln79 – remove “prevention and” 

8.     Ln86 - please add oral health before “prevention programs”

MM:

1.     An explanation is needed about why the search was narrowed down for the past five years

2.     The PRISMA flowchart (or text version of it) of the article selection is missing. Please add numerical data on eligibility screening steps. 

3.     A qualitative evaluation of the studies is not present. Please explain the quality assessment methods and the corresponding results of such tests.

4.     Please state clearly the PICO/SPIDER questions as it defines the study materials. 

5.     The inclusion and exclusion criteria within Table 1 are confusing. Please use text instead. Why the inclusion and exclusion criterium is the same for the intervention? 

6.     State how many studies you analyzed for mucositis in Ln143.

7.     Radioinduced mucositis subheadings: the representation of findings from various RCTs is better to be combined in two main groups: one with significant improvements and the other without differences compared to placebo

8.     3.3 subheadings (Ln218) should be replaced by: Prevention of postoperative infection    oral health improvement

9.     The study by Lee belongs to Radioinduced caries (under 3.4.)

10.  The study by Morais belongs to Oral mucositis (3.1.)

Discussion:

1.     Ln226 – “dental hygienist” should be changed to dental professionals. The management of oncological treatment is the responsibility of the oncology team.

2.     Ln229-231 is better to be placed in the Introduction after Ln50.

3.     Ln 232, the sentence should be corrected as “Based on the articles selected (remove “Thanks to”) from the systematic review process, we draw up a recommendation of operational protocols for dental professionals. We also included a locally used, experience gained approach from S.C ….”  

4.     Please remove “and by referring to articles dealing with the management of oral hygiene in patients with maxillary defects undergoing maxillofacial surgery” - this topic is not covered in the review

5.     Figure 2. The title should be: “Flowchart of recommended operational protocol for oral health management in HNC patients.”

6.     Please delete the sentence from the text in Ln261-262

7.     Ln262 - Delete “the main problem during radiotherapy is.” Start the sentence from “Radioinduced…”

8.     Ln264 – add “is characterized by” after “radiation damage”

9.     Continue Ln266 with the sentences from Ln287-289.

10.  Ln273-276 belong to MM (3.1 oral mucositis)

11.  Please define “Marseille soap.”

Author Response

Dear Reviewer,

I also revised the text to improve the iThenticate report.

Best regards 

Jacopo Lanzetti

Reviewer 2 Report

" Management of oral hygiene in head-neck cancer patients undergoing oncological surgery and radiotherapy: a systematic review "

It is very interesting to synthesize the available evidence for the treatment of the head and neck cancer patient undergoing resective surgery and radiotherapy with or without concomitant chemotherapy and to draft a diversified oral hygiene protocol in the various stages of cancer therapy. However, there are a few corrections that are essential to meet the standard for publication. Please refer to the following comments.

1)    Please add a Summary assessment of risk of bias for systematic reviews. This is necessary for systematic review. Also, please add a consideration for risk of bias.

2)    Please include the excluded studies in the supplementary file as well.

Author Response

(The authors gave the same response as above.)

Round 2

Reviewer 1 Report

The requested changes were completed.

Author Response

Dear reviewer,

Thank you for your comments.

Best Regards

Jacopo Lanzetti

Reviewer 2 Report

Thank you for giving me this opportunity to re-review your revised manuscript. 

Risks of bias for systematic reviews have not yet been added. Please consider adding.

Also, please clarify the purpose of this systematic review paper. The flow of this manuscript is a narrative review.

Author Response

Dear reviewer,

I corrected our manuscript, please see the attachment.

Best regards

Jacopo Lanzetti

Round 3

Reviewer 2 Report

Thank you for giving me this opportunity to re-review your revised manuscript. 

I am happy that all of the suggested corrections have been made.

Thank you for spending so much time for revised manuscript.